# Porosity Elimination in Modified Direct Laser Joining of Ti6Al4V and Thermoplastics Composites

**Haipeng Wang [1], Yang Chen [2], Zaoyang Guo [3,\*] and Yingchun Guan [1,4,5,\*]** 

[1] School of Mechanical Engineering and Automation, Beihang University, 37 Xueyuan Road, Beijing 100191, China; wang_haipeng@buaa.edu.cn

[2] School of Aeronautic Science and Engineering, Beihang University, 37 Xueyuan Road, Beijing 100191, China; chenyang_l@163.com

[3] School of Science, Harbin Institute of Technology, Shenzhen 518055, China

[4] National Engineering Laboratory of Additive Manufacturing for Large Metallic Components, Beihang University, 37 Xueyuan Road, Beijing 100191, China

[5] Hefei Innovation Research Institute, Beihang University, Hefei 230013, China

\* Correspondence: z-guo@foxmail.com (Z.G.); guanyingchun@buaa.edu.cn (Y.G.); Tel.: +86-185-1086-7097 (Z.G.); +86-010-8233-9691 (Y.G.)

**Abstract:** Hybrid lightweight components with strong and reliable bonding qualities are necessary for practical applications including in the automotive and aerospace industries. The direct laser joining method has been used to produce hybrid joints of Ti6Al4V and glass fiber reinforced polyamide (PA66-GF30). Prior to the laser joining process, a surface texturing treatment is carried out on Ti6Al4V to improve joint strength through the formation of interlock structures between Ti6Al4V and PA66-GF30. In order to reduce the generated micro-pores in Ti6Al4V-PA66-GF30 joints, a modified laser joining method has been proposed. Results show that only very few small micro-pores are generated in the joints produced by the modified laser joining method, and the fracture strength of the joints is significantly increased from 13.8 MPa to 41.5 MPa due to the elimination of micro-pores in the joints.

**Keywords:** direct laser joining; fracture strength; micro-pores; Ti6Al4V; PA66-GF30

## 1. Introduction

Hybrid component manufacturing for lightweight constructions has attracted much attention due to the increasing demands in the automotive [1], aerospace [2] and biomedical industries [3]. Hybrid components consisting of plastic and metal can reduce product weight, tailor product properties and enhance the flexibility in product design. Short glass fiber reinforced polyamide-66 is a promising candidate for lightweight construction applications due to the required lightweight and the expected thermomechanical properties [4,5]. Comparing to conventional joining methods such as adhesive bonding [6] and mechanical fastening [7], the laser joining process shows more flexibility without weight addition or contamination for environment [8]. A variety of laser-assisted joining methods have been developed for bonding plastic to metals. Rodríguez-Vidal produced joints of steel and glass fiber reinforced polyamide (PA6-GF30) by a laser-assisted metal and plastic joining (LAMP) process, and obtained a shear strength of less than 30 MPa [9]. Based on laser surface texturing and laser irradiation joining processes, Amend proposed a thermal joining method for bonding thermoplastics (PA6, PA66-GF30) to aluminum; the fracture strength of the produced Al-PA66-GF30 joints reached around 20 MPa [10].

However, laser joining process induced micro-pores inside plastic are a major concern. It has been reported that the bubbles caused by gas in plastic result in the formation of micro-pores after

solidification, which will have a negative effect on joint strength [11,12]. Zhang [13] revealed that the laser-induced shrinkage porosity in the joint of carbon fiber reinforced polymer (CFRP) and steel could be suppressed by laser fabricating protrusions on the steel surface to change the heat conduction path during the LAMP process. Chen developed an ultrasonic-aid laser joining (U-LAMP) method for direct bonding of polyethylene terephthalate (PET) and titanium, and most of the laser-induced bubbles in joints were effectively eliminated. The obtained joint strength was improved, i.e., was four times greater than that prepared by conventional LAMP method, due to the reduction of the number of bubbles in the joints [14].

In this work, the shear strength of the Ti6Al4V-PA66-GF30 joints produced by the conventional LAMP process was examined carefully. It was confirmed that the joint strength was limited by the micro-pores near the melted zone boundary of PA66-GF30. To eliminate the micro-pores to a large extent, a simple direct laser joining method has been proposed in which the clamping configuration is optimized. Results show that only very few small micro-pores are generated in the joints produced by the modified laser joining method, and that the fracture strength of the joints is significantly increased from 13.8 MPa to 41.5 MPa, while the highest value we found in the literature was only around 20 MPa. Moreover, failure mechanisms of the produced Ti6Al4V-PA66-GF30 joints have been investigated.

## 2. Materials and Methods

Commercial Ti6Al4V and PA66-GF30 were used as joining partners. PA66-GF30 is a polyamide resin (PA66) matrix reinforced by 30% chopped glass fibers. Ti6Al4V and PA66-GF30 specimens have a thickness of 3 mm and 5 mm, respectively. Both Ti6Al4V and PA66-GF30 specimens have dimensions of 20 mm in width, and 70 mm in length. Before laser joining PA66-GF30 to metal, surface texturing was performed on Ti6Al4V to improve the adhesion between PA66-GF30 and Ti6Al4V. Laser surface texturing was carried out with a continuous-wave (CW) fiber laser, and laser joining of PA66-GF30 and Ti6Al4V sheets were conducted with a nanosecond laser with central wavelength of 1064 nm. The modified laser processing parameters used in the experiments are shown in Table 1.

**Table 1.** Laser parameters used in microstructure fabrication and laser joining process.

| Process | Laser Power (W) | Frequency (kHz) | Pulse Duration (ns) | Scan Speed (mm s$^{-1}$) |
|---|---|---|---|---|
| Texturing | 75 | CW mode | - | 100 |
| LAMP | 55 | 100 | 90 | 1000 |

A total surface area of 5 mm × 20 mm was textured on Ti6Al4V and overlapped with PA66-GF30 sheets. Figure 1a illustrates the schematic diagram of the commonly-used LAMP method. The Ti6Al4V specimen was placed on a PA66-GF30 sheet, and two preset pads were used to set the level of two connected parts. Two preset clamping Ti6Al4V plates with thickness of 1.5 mm were used to provide suitable pressure for the overlapping area, as shown in Figure 1a. Laser irradiates the Ti6Al4V plate surface and heat conducts from the plate surface to the Ti6Al4V-PA66-GF30 interface. The plastic near the interface melts and flows into the created grooves on Ti6Al4V surface under the clamping pressure. To improve the joint strength, a modified laser joining process, as schematically illustrated in Figure 1b, was proposed to eliminate the micro-pores generated in LAMP produced joints. The preset pads were removed, and a preset clamping force was exerted on each side of the overlap area during the laser joining process. Structure density is defined as the ratio of the structured area to the overall overlapped area, and can be varied by controlling the pitch of laser scans.

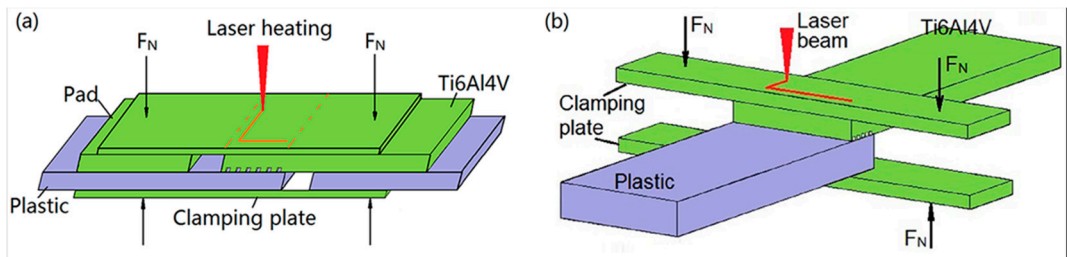

**Figure 1.** Schematic diagrams of laser joining process. (**a**) Conventional direct laser joining process; (**b**) the modified laser joining process.

After the laser joining process, tensile shear tests were carried out to evaluate the fracture strength of the produced joints. The schematic representation of the shear tests is exhibited in Figure 2. Two pads of the same thickness as the PA66-GF30 and Ti6Al4V parts were used for alignment during the tests. The tests were conducted at a travelling speed of 2 mm/min by using Instron universal testing machine (Instron 5982) with the maximum load of 30 kN. Cross-sections of the produced joints were characterized using optical microscopy and scanning electronic microscopy (SEM, JSM-6701F) equipped with energy dispersive X-ray spectroscopy (EDS).

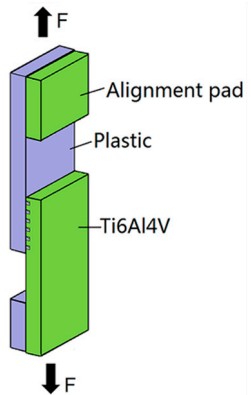

**Figure 2.** Schematic representation of tensile shear test.

## 3. Results and Discussion

### 3.1. LAMP Method

Groove structures with a depth of 350 μm were fabricated on Ti6Al4V surface, as shown in Figure 3a. The width of the grooves is around 300 μm (Figure 3b). Grooves with structure density ranging from 25% to 75% have been fabricated on Ti6Al4V by changing the pitch of grooves (Figure S1). Figure 3c shows the images of LAMP joined Ti6Al4V-PA66-GF30 specimens. Cross-section morphology of the joint is exhibited in Figure 3d. It can be observed that the grooves have been completely filled by the melted PA66-GF30 during LAMP process and the interlock structure forms between Ti6Al4V and PA66-GF30 after solidification of PA66-GF30 melts. In the joint, a large number of micro-pores have been observed in the interior of PA66-GF30 with sizes ranging from dozens to hundreds of microns. These micro-pores can be divided into two types on basis of morphology and distribution [15]. One type is generated close to the bonding interface or located in the interior of PA66-GF30 filled in the grooves, where the materials experience very high temperatures during laser joining process [13,15]. These pores independently distribute with irregular shapes and the size of over 100 μm, as indicated in Figure 3d. These pores most likely result from the pyrolysis of PA66 due to the high temperature [16]. The pyrolysis of PA66 produces a large number of gaseous products such as ammonia and carbon dioxide. The gaseous products inside the melts result in the generation of micro-pores in the interior of PA66-GF30 after solidification of the melts.

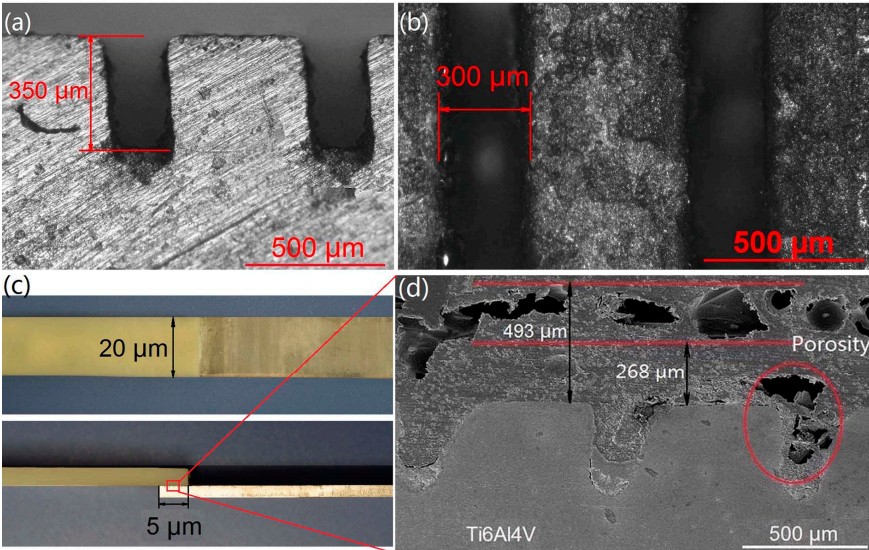

**Figure 3.** (**a**,**b**) Laser created groove structures on Ti6Al4V surface with depth of around 350 μm (**a**) and width of around 300 μm (**b**); (**c**) images of LAMP produced Ti6Al4V-PA66-GF30 specimen; (**d**) cross-section of LAMP produced Ti6Al4V-PA66-GF30 joint.

The other type of pores distributes close to the boundary of melted zone and has a distance of around 200-500 μm from Ti6Al4V-PA66-GF30 bonding interface [15], as indicated in Figure 3d and Figure S2. These pores exhibit rough morphologies and arbitrary shapes, and distribute throughout the cross-section regardless of the structure density, as can be seen in Figure S2. The size of these pores ranges from dozens to hundreds of microns, as shown in Figure 3d. These pores are mainly caused by the shrinkage of PA66-GF30 melts during solidification. Shrinkage porosity is commonly formed in the manufacture of composite polymeric materials due to solidification shrinkage [17,18]. Ti6Al4V has much higher thermal conductivity (7 W/(m K)) than PA66-GF30 (0.30 W/(m K)) [19,20]. After the laser joining process, the heat quickly conducts by Ti6Al4V, and the PA66-GF30 close to the bonding interface solidifies first due to the higher thermal conductivity of Ti6Al4V compared to that of PA66-GF30. The PA66-GF30 melts near the boundary of melted zone solidify last, and solidification shrinkage of PA66-GF30 results in the formation of porosity near the melted zone boundary [15].

Structure density is a key parameter that determines the effective bearing area of the joint interface during tensile shear test, and it thus significantly influences the joints strength [21]. The grooves' structure, with depths of around 350 μm, was fabricated on the Ti6Al4V surface to investigate the bonding strength of the Ti6Al4V-PA66-GF30 joints produced through the laser joining process. The results of the tensile shear tests (Figure 4a) indicate that the fracture strength of the joints increases from around 8.8 MPa to 13.8 MPa as the structure density increases from 25% to 58%, which is ascribed to increasing the effective bearing area in joint interface. When the structure density reaches 58%, the fracture strength of the joints does not increase anymore.

Figure 4b–d shows the surface morphologies and cross-sections of Ti6Al4V-PA66-GF30 joints with different structure density after tensile shear tests. In the case of a structure density of 25%, PA66-GF30 was pulled out from micro-grooves during tensile tests, while little plastic debris remained in the micro-grooves, as shown in Figure 4b. As the structure density increased to 42%, part of PA66-GF30 filled in the micro-grooves was pulled out, and the majority of PA66-GF30 remained in the micro-grooves or adhered to the Ti6Al4V surface, as shown in Figure 4c. In these cases, the joints failed at the bonding interface between the Ti6Al4V and PA66-GF30. Because the debonding stress between the untreated Ti6Al4V surface and PA66-GF30 is much lower than the shear strength of the PA66-GF30 material, the shear strength of the joints depends on the size of the structured area, i.e., the structure density. However, when the structure density increased to 58% and above, the shear strength of the bonding surface became stronger than that of the porous zone near the melted zone boundary. In these

cases, the joints failed by the fracture of PA66-GF30 near the melted zone boundary of PA66-GF30. As exhibited in Figure 4d–e, a layer of PA66-GF30 with a thickness of around 370 μm was separated from the matrix after a tensile shear test, as well as a layer of PA66-GF30 with a thickness of less than 370 μm remaining on the Ti6Al4V surface. Compared to the porosity distribution in the joint as displayed in Figure 3d, we can see that the fracture of the joints occurred in the porous zone inside PA66-GF30 and close to the boundary of melted zone. The generated pores close to the boundary of the melted zone limit the bonding strength of Ti6Al4V-PA66-GF30 joints. Therefore, when the tensile load reaches the fracture strength of PA66-GF30 in the porous zone, the joints will fail regardless of the structure density on the Ti6Al4V surface.

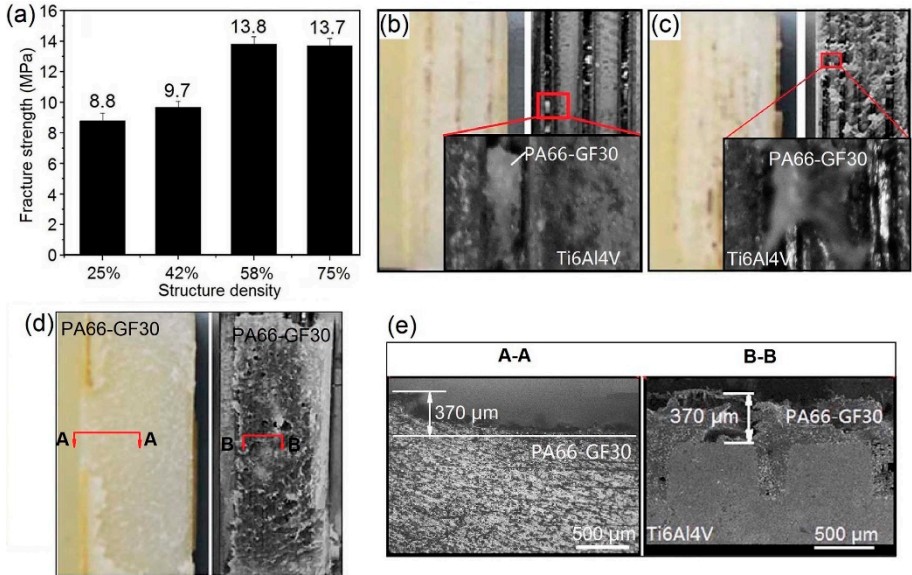

**Figure 4.** (**a**) Fracture strength of LAMP produced Ti6Al4V-PA66-GF30 joints with various structure densities on Ti6Al4V surfaces; (**b**)–(**d**) surface morphologies and cross-section morphologies of LAMP produced Ti6Al4V-PA66-GF30 joints with structure density of 25% (**b**), 42% (**c**), and 58% (**d**), respectively; (**e**) cross-sections of the Ti6Al4V and PA66-GF30 parts of the failed Ti6Al4V-PA66-GF30 joint in (**d**).

### 3.2. Modified Laser Joining Method

In order to improve the bonding strength of Ti6Al4V-PA66-GF30 joints, a modified laser joining process has been proposed, as schematically illustrated in Figure 1b, to eliminate the micro-pores generated in joints. In this process, the preset pads in Figure 1a are removed and a preset clamping force is applied on the overlapping area during laser joining process, so that the excess melted PA66-GF30 overflows from the micro-grooves and the formed micro-pores derived from the pyrolysis of PA66 are expelled out with the flow of PA66-GF30 melts. As the preset clamping force is always applied on the overlapping area during the laser joining process, the solidification shrinkage-induced pores near the boundary of melted zone (Figure 1a) can be reduced significantly, as shown in Figure S3. From Figure 5a, it can be observed that no large size pores are generated in the joint, and very few small pores with sizes of around 30 μm are observed (see inset in Figure 5a). Figure 5b shows the fracture strength of the produced Ti6Al4V-PA66-GF30 joints as a function of the structure density on Ti6Al4V surface. The fracture strength of the produced joints monotonically increases with increasing structure density, and reaches up to around 41.5 MPa as the structure density increases to 75%. This fracture strength is much higher than the reported joints produced between cyclic olefin polymer (COP)/PA6.6-GF35 and steel (<22 MPa) [22,23], PA66-GF30/Acrylonitrile butadiene styrene (ABS) and aluminum (<22 MPa) [9], PET and titanium (<20 MPa) [14,24] by the laser joining process.

All the produced Ti6Al4V-PA66-GF30 joints by the modified laser joining process fractured at the bonding interface between Ti6Al4V and PA66-GF30 during tensile shear tests. Figure 5c shows

fracture morphology of the joints after tensile shear test, which suggests that the fracture results in a smooth and neat fracture interface. Therefore, as the structure density on Ti6Al4V increases, the fracture strength of the joints monotonically enhances, due to the increasing effective bearing area. The modified laser joining process has been demonstrated to be a feasible and reliable method to eliminate the pores generated in the joints. The elimination of large quantities of pores inside PA66-GF30 leads to significant enhancement of the joint strength, as shown in Figures 3–5.

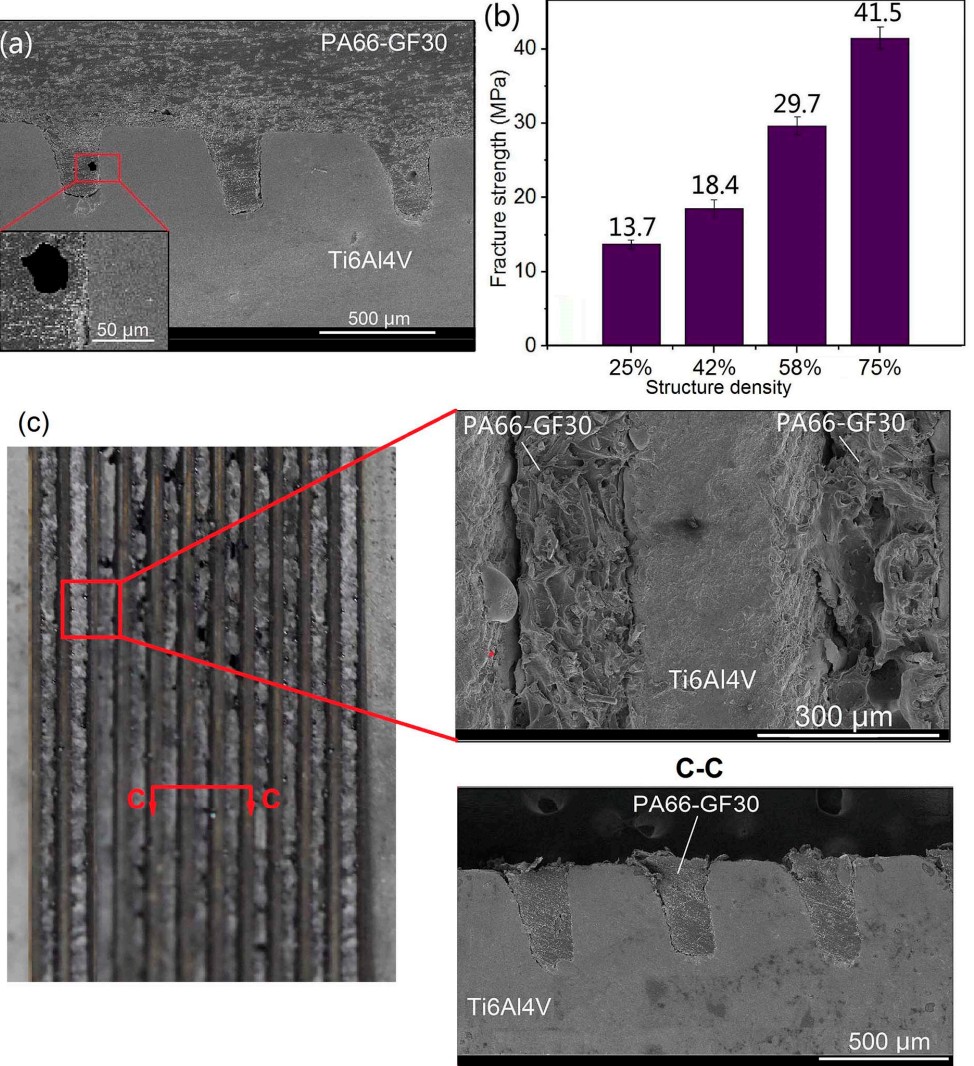

**Figure 5.** Cross-sections and fracture strength of Ti6Al4V-PA66-GF30 joints produced by modified laser joining process. (**a**) Cross-section of the produced Ti6Al4V-PA66-GF30 joints; (**b**) fracture strength of Ti6Al4V-PA66-GF30 joints as a function of structure density; (**c**) surface and cross morphologies of Ti6Al4V-PA66-GF30 joint after tensile shear test.

## 4. Conclusions

A direct laser joining process was carried out to bond thermoplastics to Ti6Al4V with a nanosecond pulse laser. Surface texturing treatment was performed on Ti6Al4V before laser joining process, producing a strong mechanical anchor effect between Ti6Al4V and PA66-GF30 after the solidification of PA66-GF30 melts. By using a conventional LAMP process, a large number of micro-pores with sizes ranging from dozens to hundreds of microns appeared in Ti6Al4V-PA66-GF30 joints due to the solidification shrinkage of PA66-GF30 and the pyrolysis of PA66 during laser joining process. These micro-pores limited the fracture strength of Ti6Al4V-PA66-GF30 joints to around 13.8 MPa,

and fractures of the joints occurred in the porous zone of PA66-GF30. A modified direct laser joining process was proposed to effectively eliminate the generated micro-pores in joints. Therefore, the fracture strength of Ti6Al4V-PA66-GF30 joints was improved to 41.5 MPa, and fractures of the joints occurred at the bonding interface between Ti6Al4V and PA66-GF30.

**Supplementary Materials:** The following are available online at http://www.mdpi.com/2076-3417/9/3/411/s1, Figure S1: The created grooves on Ti6Al4V surface with structure density of 25% (**a**), 42% (**b**), 58% (**c**), and 75% (**d**), respectively; Figure S2: Cross-sections of LAMP produced joints with structure density of 25% (**a**), 42% (**b**), 58% (**c**), and 75% (**d**), respectively; Figure S3: Cross-sections of the modified laser joining method produced Ti6Al4V-PA66-GF30 joints with structure density of 25% (**a**), 42% (**b**), 58% (**c**), and 75% (**d**), respectively.

**Author Contributions:** H.W., Y.G. and Z.G. conceived and designed the experiments; H.W. performed the experiments; H.W. and Y.C. analyzed the data; H.W. and Y.G. wrote the paper; Z.G. contributed materials.

**Funding:** This work was supported by Beijing Natural Science Foundation under Grant J170002; National Natural Science Foundation of China under Grant 51705013; National Key Research and Development Program of China under Grant 2018YFB1107400, 2018YFB1107700, and 2016YFB1102503; National Key Basic Research Program of China under Grant 2015CB059900.

**Acknowledgments:** The authors thank Hongyu Zheng and Wenhe Feng from Singapore Institute of Manufacturing Technology for their advice and assistance during the research.

**Conflicts of Interest:** The authors declare no conflict of interest.

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
