# Peer review of "Porosity Elimination in Modified Direct Laser Joining of Ti6Al4V and Thermoplastics Composites"

_applsci, doi:10.3390/app9030411_

Reviewer 1 Report

The originality of the presented is rather low. Micro-texturing using laser technology in order to improve the adhesive bonding has studied extensively much earlier than the reference paper by Zhang(ref. 13) for instance the paper by RCY Wong in the Journal for Materials Processing Technology published in 1997 volume. 67. Here of course, the situation is slightly different as it concerns direct bonding of dissimilar materials. From the introduction one gets the impression that the originality of this work is limited to the change of the clamping configuration. This should be extended and better described. What is a large pre-set clamping force (page 2, line 75) and how critical is this value? Was there a particular reason for doing the testing not in agreement with some standards (like ASTM D1002)? Figure S2 and S3 seems to be identical?

with respect to the English language and style: the text should be corrected! Please pay special attention to the use of the articles 'the', 'a' and 'an'.

Author Response

Reply to Reviewer’s Comments Applied Sciences Title: Porosity Elimination in Modified Direct Laser Joining of Ti6Al4V and Thermoplastics Composites Authors: Haipeng Wang, Yang Chen, Zaoyang Guo, and Yingchun Guan We are very grateful to the reviewers for the valuable comments. We have carefully considered the comments and made modifications in the manuscript accordingly as listed below. All the modifications are shown in BLUE in the revised manuscript. Reviewers' comments: Reviewer #1: The originality of the presented is rather low. Micro-texturing using laser technology in order to improve the adhesive bonding has studied extensively much earlier than the reference paper by Zhang (ref. 13) for instance the paper by RCY Wong in the Journal for Materials Processing Technology published in 1997 volume. 67. Here of course, the situation is slightly different as it concerns direct bonding of dissimilar materials. Reply: It is true that micro-texturing using laser technology in order to improve the adhesive bonding has been studied extensively much earlier than our work. The originality of our manuscript is that the modified LAMP method we proposed can improve the quality of the bonding significantly. As shown in our manuscript, the fracture strength of the joints has been increased from 13.8MPa to 41.5 MPa, while the highest record we found in the literature is only around 20 MPa. We believe this huge improvement will promote the applications of the direct laser joining method in various engineering fields. 1. -Introduction: From the introduction one gets the impression that the originality of this work is limited to the change of the clamping configuration. This should be extended and better described. Reply: According to Reviewer’s suggestion, the introduction part has been revised to highlight the originality of this manuscript in Line 52-59 on Page 2 as follows: “In this work, the shear strength of the Ti6Al4V-PA66-GF30 joints produced by the conventional LAMP process was examined carefully. It was confirmed that the joint strength was limited by the micro-pores near the melted zone boundary of PA66-GF30. To eliminate the micro-pores to a large extend, a simple direct laser joining method has been proposed by optimizing the clamping configuration. Results show that only very few small micro-pores are generated in the joints produced by the modified laser joining method, and the fracture strength of the joints is significantly increased from 13.8 MPa to 41.5 MPa, while the highest record we found in the literature is only around 20 MPa. Moreover, failure mechanisms of the produced Ti6Al4V-PA66-GF30 joints have been investigated.” 2. - Materials and Methods: What is a large pre-set clamping force (page 2, line 75) and how critical is this value? Reply: The clamping force is critical to the quality of the joints. Using classical LAMP method, the applied clamping force is not able to be transferred to the overlapping area because the existence of the preset pads. However, in the modified laser joining process, the clamping force has been transferred to the overlapping area effectively. As pointed out in Section 3.2, this is essential to the elimination of the micro-pores. 3. Was there a particular reason for doing the testing not in agreement with some standards (like ASTM D1002)? Reply: In this study, tensile tests have been performed using two extra pads. The main reason is for the alignment of clamping positions of upper and lower clamping devices, as shown in the following Figure. Figure. 2 Schematic representation of tensile shear test used in this work. 4. Figure S2 and S3 seems to be identical? Reply: Figure S2 and S3 shows cross-sections of the joints by LAMP method and modified LAMP method, respectively, while the main difference between Figure S2 and S3 lies in the distribution of the formed micro-pores in the cross-sections of joints. In LAMP produced joints, a large amount of micro-pores with rough morphologies and arbitrary shapes has been found throughout the cross-sections (Figure S2). Whereas, in the modified LAMP produced joints, no obvious micro-pores can be observed (Figure S3). 5. With respect to the English language and style: the text should be corrected! Please pay special attention to the use of the articles 'the', 'a' and 'an'. Reply: According to Reviewer’s suggestion, the text has been proofread several times by the authors, and the use of the articles 'the', 'a' and 'an' has been fixed.

Reviewer 2 Report

The manuscript is well written and the results presented are justified. 

Comments regarding Fig 1: Both (a) and (b) figures are showing FN top clamping plates through which laser beam is going through and heating the Ti6Al4V substrates. Is this correct? I think the figures misleads and need to be corrected.

Line 129 - 131: why fracture strength  of the joint does not increase after 58% structure density? please add a sentence in this paragraph.

Author Response

Reply to Reviewer’s Comments

Applied Sciences

 Title: Porosity Elimination in Modified Direct Laser Joining of Ti6Al4V and Thermoplastics Composites

Authors: Haipeng Wang, Yang Chen, Zaoyang Guo, and Yingchun Guan

We are very grateful to the reviewers for the valuable comments. We have carefully considered the comments and made modifications in the manuscript accordingly as listed below. All the modifications are shown in BLUE in the revised manuscript.

Reviewers' comments:

Reviewer #2: The manuscript is well written and the results presented are justified.

Reply: Thanks for the reviewer’s positive acknowledgement of the novelty and the language of this work.

1. Comments regarding Fig 1: Both (a) and (b) figures are showing FN top clamping plates through which laser beam is going through and heating the Ti6Al4V substrates. Is this correct? I think the figures misleads and need to be corrected.

Reply: Schematic diagrams in Figures 1(a) and 1(b) showing laser joining process are correct. In both 1(a) and 1(b) figures, laser irradiates on the top surface of clamping plate (Ti6Al4V), and heat conducts from Ti6Al4V plate surface to Ti6Al4V-PA66-GF30 interface. Thus, the accumulated energy at Ti6Al4V-PA66-GF30 interface heats PA66-GF30 surface.

2. Line 129 - 131: why fracture strength of the joint does not increase after 58% structure density? please add a sentence in this paragraph.

Reply: According to Reviewer’s suggestion, it is explained in Line 143-149 on Page 4 as follows: “In these cases, the joints failed at the bonding interface between the Ti6Al4V and PA66-GF30. Because the debonding stress between the untreated Ti6Al4V surface and PA66-GF30 is much lower than the shear strength of the PA66-GF30 material, the shear strength of the joints depends on the size of the structured area, i.e., the structure density. However, when structure density increased to 58% and above, the shear strength of the bonding surface became stronger than that of the porous zone near the melted zone boundary. In these cases, the joints failed by the fracture of PA66-GF30 near the melted zone boundary of PA66-GF30.”
